# Highly ordered macroporous hydrogen-bonded organic frameworks based on small biocompatible molecules

Qiu-Xia Li[1,2,3], Wan-Zhen Cai[1], Xiao-Liang Ye[1], Yi Zeng[4], A. R. Mahammed Shaheer[1], Zai-Sheng Ye[4] & Tian-Fu Liu [1,2,3] ✉

Template method offers a promising strategy for synthesizing large pore inaccessible through traditional molecular design. However, this approach has not yet been successfully implemented in molecular assemblies based on weak non-covalent interactions (NCIs), mainly because the assemblies often deviate from original structures during template-assisted syntheses, and the resulting porous structures lack the robustness to survive upon template removal. In this work, we address these challenges through choosing small biocompatible building blocks featuring multiple hydrogen-bonded sites and extensive π conjugation, enabling self-assembly into desired structure in the presence of templates and ensure structural integration upon template removal. As a result, the transformation from densely packed hydrogen-bonded crystalline materials to macroporous structure, referred to as hydrogen-bonded organic frameworks (HOFs), becomes achievable. This strategy facilitates the fabrication of highly ordered materials in single-crystal form with high physiological stability, and enhanced mass transfer. Importantly, it greatly broadens the HOF library to small, affordable, low-toxic, and clinically applicable molecules, making HOFs promising biocompatible porous substrates for bio-related applications such as enzyme immobilization. Herein, we successfully loaded trypsin into macroporous HOFs, which function as effective cellular scaffolds and promote the differentiation of peripheral blood mononuclear cells into fibrocytes, demonstrating their promising potential for biologic applications.

Porous materials constructed from molecular building blocks have attracted great attention due to their wide-range applications in catalysis, separation, and medicine[1-4]. Depending on specific application, the necessary pore size varies from micropore to macropore. For molecular-based porous materials, pore size ranging from a few angstroms to a few nanometers is relatively feasible through judiciously designing the building blocks[5-8]. However, beyond this size would bring significant challenges in terms of organic synthesis, which would not only increase the costs but also pose risks of high toxicity to living organisms and instability in bio-related applications[9-12]. Especially, achieving macropores or pores with specific shapes is extremely challenging, at times, unachievable.

To address these challenges, template-assisted method—in which a predefined template guides the formation of large pores—has been applied for fabricating meso or macroporous molecular-based materials such as organic polymer, metal-organic frameworks, and covalent organic frameworks[13-17]. However, this method has yet to be applied to the molecular assemblies based on weak intermolecular interactions.

[1]State Key Laboratory of Structural Chemistry, Fujian Institute of Research on the Structure of Matter, Chinese Academy of Sciences, Fuzhou, Fujian, China. [2]University of the Chinese Academy of Sciences, Beijing, China. [3]Fujian College, University of Chinese Academy of Sciences, Fuzhou, China. [4]Department of Gastric Surgical Oncology, Clinical Oncology School of Fujian Medical University, Fujian Cancer Hospital, Fuzhou, Fujian, China. ✉e-mail: tfliu@fjirsm.ac.cn

The reason lies in that: (1) most non-covalent interactions (NCIs) between molecules are relatively weak and often lack of strong directionality[18,19]. As a result, the presence of a template usually disrupts the self-assembly process, leading to materials with large pores but unexpected or unknown structures. (2) Structures formed through weak NCIs are typically too fragile to withstand the removal of the template, which prevents the formation of stable porous structures[20,21]. For these reasons, current strategies for fabricating macroporous materials based on NCI assembly, such as hydrogen-bonded organic frameworks (HOFs), have largely relied on crystal morphology control to create void spaces[22–24]. For example, tuning solvent volatilization rates induces rapid anisotropic self-assembly to generate macropores. However, a broadly applicable strategy for producing HOFs with high-density and uniformly macropores has yet to be established. Addressing above-mentioned issues would enable the fabrication of macroporous HOFs with tunable pore sizes, excellent biocompatibility, and high crystallinity. Such materials would hold great promise for applications in the biomedical field and beyond[25–28].

In this study, we carefully selected a series of small biocompatible molecular building blocks with multiple hydrogen donor/acceptor sites and extensive π-conjugation. The multiple hydrogen bonds and extensive π−π interaction between molecules not only effectively guide the self-assembly to the expected structure in the presence of templates, but also ensure the stability of assemblies upon template removal. As a result, highly ordered macroporous HOFs in single-crystal form (denoted by OM-HOFs) were successfully synthesized through template method, and its structure can be unambiguously determined through electron-diffraction analysis. Such a type of macroporous material based on biocompatible molecules emerge as promising candidates for enzyme immobilization with enhanced stability and recyclability. Our study demonstrated that Try@OM-PFC, employed as a cellular scaffold, can promote fibrocyte differentiation, offering the advantages of transparency for observation, shape adaptability, and moisture retention.

## Results and discussion
### Syntheses and structural characterizations of OM-HOF single crystals based on small biocompatible molecules
Melamine (MA) and tricarboxylic acid (TMA) were selected as building blocks to self-assemble a previously reported co-crystal through hydrogen bonds (Fig. 1 and Supplementary Fig. 1)[29]. In the crystal structure, four MA were connected with two TMA to form a two-dimensional layer. Abundant hydrogen bonds between building blocks and the π−π stacking between adjacent layers were expected to impart high structural stability even after creating void space. Highly ordered polystyrene (PS) monoliths with an average diameter of 200 nm were used as hard template upon crystallization and then were removed to generate an interconnected and highly ordered macroporous structure named as OM-PFC21 (Fig. 2a, PFC represents Porous Frameworks from Chinese Academy of Sciences). The identical structures of OM-PFC21 and MA-TMA co-crystal were confirmed by their consistent powder X-ray diffraction (PXRD) patterns (Supplementary Fig. 3a).

The method can be further extended into other small biocompatible or clinically applicable molecules. Following the same procedure, OM-PFC22 based on m-trihydroxybenzene (THB) and TMA, and OM-PFC23 based on methylene blue (MB) and uric acid (UA) were also successfully synthesized, which were confirmed by their consistent PXRD patterns (Supplementary Fig. 3b, c). Scanning electron microscopy (SEM) images clearly revealed that OM-PFC21, 22 and 23 inherit the morphology of ordered template (Fig. 2a–c). Thus, well-arranged spherical macropores showed a diameter of ca. 200 nm and were interconnected by the 30-60 nm channel (Supplementary Fig. 9). The porosity of the activated macroporous HOFs was investigated by nitrogen adsorption measurements at 77 K. Supplementary Fig. 4 showed type IV isotherms with a significant hysteresis loop and steeply increased adsorption in the high-pressure region for these materials, suggesting the existence of macropores in structures. Mercury intrusion porosimetry measurements (Fig. 2d–f), which can be used to assess the pore size larger than 3 nm, were applied to verify cumulative pore volume and indicate the pore size of about 60 nm for OM-PFC21, 22, and 23. As expected, all these macroporous materials can maintain structure integrity upon the removal of the template due to abundant NCIs in structures.

Very strikingly, selected-area electron diffraction patterns (Fig. 2g, h) exhibited clear crystal lattices, revealing the single-crystalline nature of macroporous HOFs, which is unprecedented in previous reports. Taking OM-PFC-23 as an example, the observed lattice planes are in accordance with the predicted ones based on x-ray crystallography (Supplementary Fig. 12), further proving the successful synthesis of OM-PFC23 crystals. Furthermore, the size of macropores can be easily

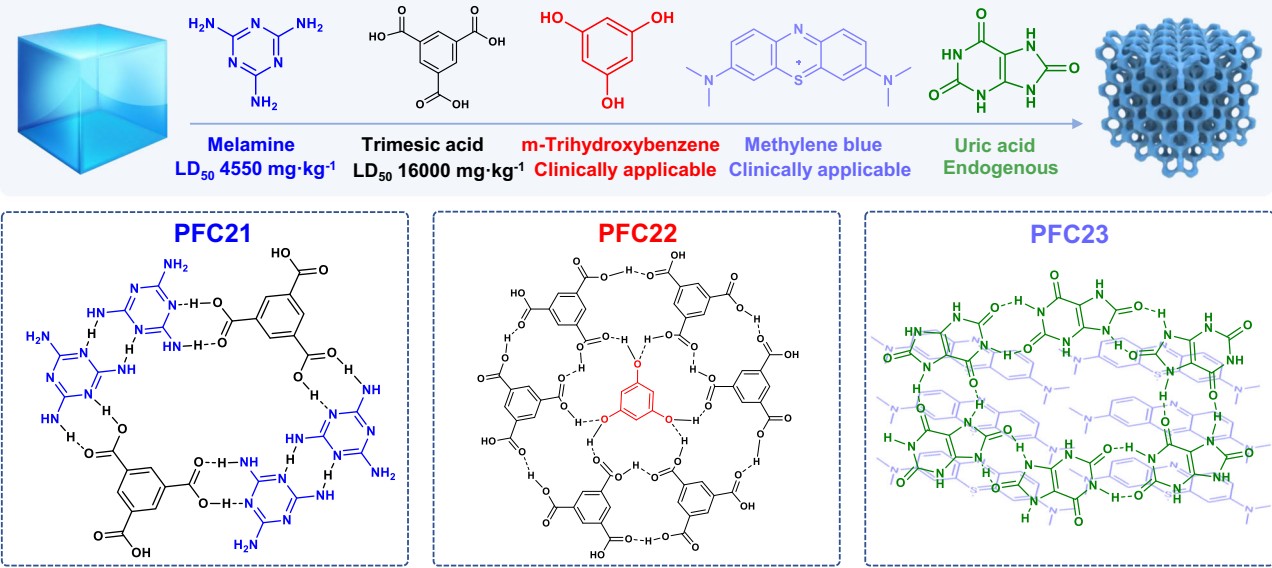

**Fig. 1** | Schematic illustration of the self-assembly process of OM-HOFs (OM-PFC21, OM-PFC22, and OM-PFC23).

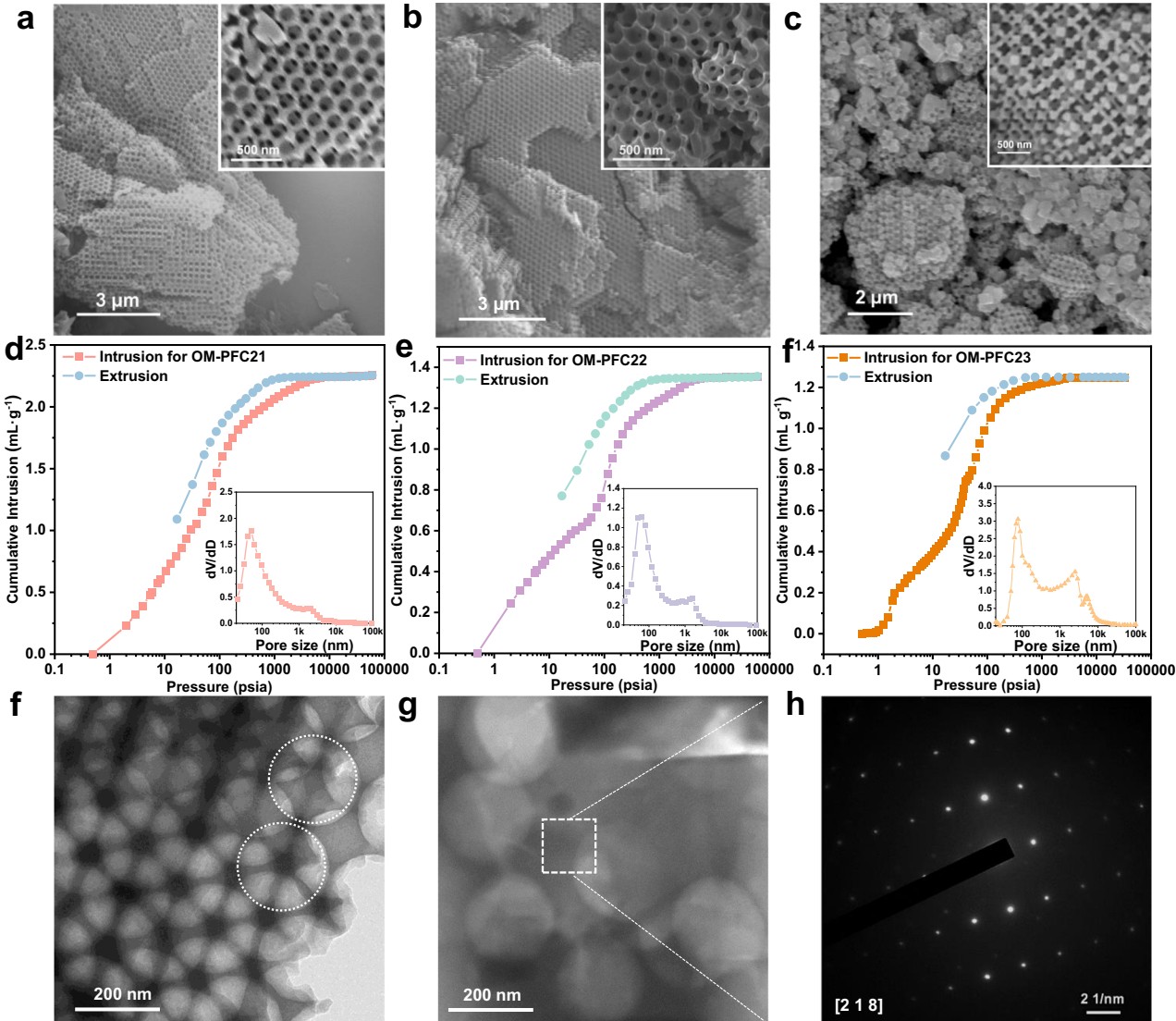

**Fig. 2 | Structural characterizations of the OM-PFC.** Scanning electron microscopy (SEM) images of **a** OM-PFC21 (Scale bar: 3 μm), **b** OM-PFC22 (Scale bar: 3 μm), and **c** OM-PFC23 (Scale bar: 2 μm), inset scale bar: 500 nm. Intrusion and extrusion branches of **d** OM-PFC21, **e** OM-PFC22, and **f** OM-PFC23 measured by mercury intrusion porosimetry, insets show the macropore size distribution (vertical axis units: $10^{-3}$ mL g$^{-1}$ nm$^{-1}$). **f, g** The transmission electron microscopy (TEM) images of OM-PFC23 (Scale bar: 200 nm), and **h** the corresponding selected-area electron diffraction (SAED) patterns (Scale bar: 2 1/nm). Three independent experiments were repeated with similar results.

tuned from 100 nm to 200 nm through varying the diameters of PSs, which were demonstrated in Supplementary Fig. 11.

## Trypsin immobilization and kinetic studies

Despite of the very high porosity, the resulting macroporous HOFs showed excellent stability in biophysical environments such as phosphate buffer solution (PBS) and artificial sweat (Supplementary Figs. 13 and 14). Therefore, such a type of low-toxic, metal-free, stable macroporous material presents as promising candidates for enzyme immobilization. Trypsin is a proteolytic enzyme widely used in many industrial and research applications, which can specifically cleave bonds at the carboxyl side of lysine (Lys) and arginine (Arg) within proteins[30]. However, this characteristic renders its susceptible to autolysis when recognizing its own Lys and Arg in solution, resulting in the formation of tryptase or even smaller fragments (Fig. 3a)[31–33]. The immobilization of trypsin would not only avoid its autolysis but also provide a fast and convenient method for its separation from insolution protein digestion reaction during practical applications[34,35]. Considering the existence of interconnected and ordered macropores

in the structure, trypsin was immobilized into OM-PFC21 and 22 to produce Try@OM-PFC composites (Fig. 3b). Confocal laser scanning microscopy images confirmed the even loading of trypsin inside OM-PFC particles (Fig. 3c). As shown in Supplementary Fig. 15, Try@OM-PFC21 and 22 retained their crystallinity after trypsin loading. Fourier transform infrared (FT-IR) spectra (Supplementary Fig. 16) show the characteristic peaks of trypsin at 1639 cm$^{-1}$ (amide I) and 1513 cm$^{-1}$ (amide II) for -CONH- group. The peak of OM-PFC21 was red-shifted from 1663 cm$^{-1}$ to 1648 cm$^{-1}$, and a shoulder nearby amide II peak was observed after immobilization, indicating the existence of intermolecular interactions between OM-HOFs and trypsin[36,37]. The similar phenomena were also observed in Try@OM-PFC22 composite. These results proved the successful loading of Trypsin in OM-HOFs.

The loading amounts were determined to be 1.21 mg mg$^{-1}$ for OM-PFC21 and 1.47 mg mg$^{-1}$ for Try@OM-PFC22, through monitoring the concentration change of solution by UV-vis (Supplementary Fig. 18 and Table 2), which were much higher than the reported adsorbents under similar conditions (Supplementary Fig. 19). To demonstrate that immobilization in a confined space does not diminish their hydrolytic

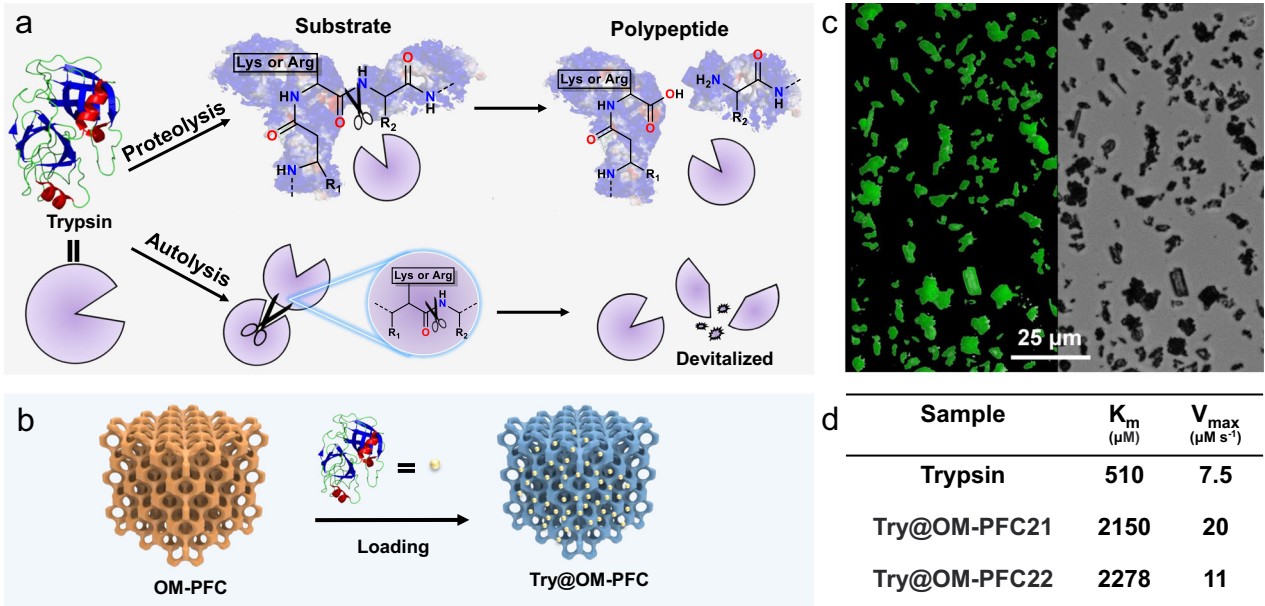

| Sample | $K_m$ (μM) | $V_{max}$ (μM s$^{-1}$) |
|---|---|---|
| Trypsin | 510 | 7.5 |
| Try@OM-PFC21 | 2150 | 20 |
| Try@OM-PFC22 | 2278 | 11 |

**Fig. 3 | Trypsin immobilization and kinetic studies. a** Schematic diagram of trypsin proteolysis substrate and autolysis process. **b** The process of trypsin immobilization within OM-PFC. **c** Confocal laser scanning microscopy (CLSM) images of OM-PFC particles loaded with fluorescein-labeled trypsin (3 independent experiments were repeated with similar results). Scale bar: 25 μm. **d** Michaelis-Menten kinetics of trypsin, Try@OM-PFC21, and Try@OM-PFC22.

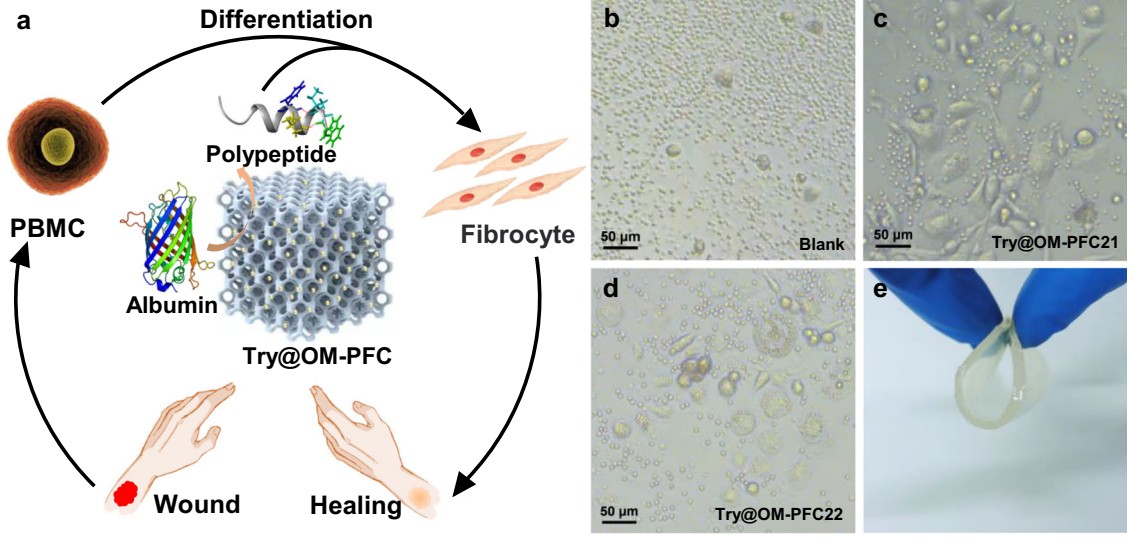

**Fig. 4 | Try@OM-PFC promoted fibrocyte differentiation. a** Schematic illustration of Try@OM-PFC potentiating fibrocyte differentiation; **b-d** peripheral blood mononuclear cells (PBMC) were cultured in 96-well plates for 9 days in the medium without (**b**) and with Try@OM-PFC21 (**c**) and Try@OM-PFC22 (**d**) treatments (three independent experiments were repeated with similar results). Scale bar: 50 μm. **e** Photograph of the viscoelastic Try@OM-PFC/GEL.

capabilities, enzymatic activities of the Try@OM-PFC21 and 22 were assessed using N-Benzoyl-Arginine-4-nitroanilide (BAPNA) as substrate. The results showed that both free trypsin and Try@OM-PFC followed Michaelis-Menten kinetics. For Try@OM-PFC21, the $V_{max}$ was calculated as 20 μM s$^{-1}$, which was around 2.5 times higher than free trypsin (7.5 μM s$^{-1}$), and the $K_m$ reached 2150 μM, which was four times higher than free trypsin (510 μM) (Fig. 4d and Supplementary Figs. 21 and 22). The $V_{max}$ and $K_m$ for Try@OM-PFC22 (Supplementary Fig. 23) were calculated as 11 μM s$^{-1}$ and 2178 μM, which were 1.5 and 4.5 times higher than free trypsin, respectively. According to these results, the following conclusions can be drawn: 1. Immobilization of enzyme in macroporous HOFs does not hinder the catalytic active site or diminish its functionality. 2. Both Try@OM-PFC21 and Try@OM-PFC22

exhibited higher $V_{max}$ than the free trypsin. $V_{max}$ is defined as the maximum reaction rate when all of the active binding sites of enzyme molecules are fully occupied with a specific substrate molecule under a given condition. We suspect that loading trypsin inside OM-HOF particles can reduce the self-aggregation and protect enzyme from inactivation or denaturation, therefore increasing the active binding sites in relative to the situation in solution. Additionally, the intermolecular interactions between substrate and OM-HOFs may facilitate the substrate diffusion and cause more efficient hydrolysis. 3. The $K_m$ values for Try@OM-PFC were higher than that of free trypsin. Since $K_m$ is an indicator of the enzyme's affinity toward specific substrates, we can infer that enzyme-substrate affinity is decreased after immobilization. The lower affinity could be attributed to the conformational rigidity

and steric effects of enzyme in a confined space. Similar phenomena have been reported in previous studies on enzyme immobilization[34,38].

## Storage stability and reusability of Try@OM-PFC

The storage stability and reusability of enzymes are important parameters for their practical applications, especially from the economical perspective. To investigate how the immobilization of enzymes in a confined space affected their stability and reusability, both free and immobilized trypsin were kept in an environment close to body temperature (37 °C) for 5 days, and their enzymatic activity were monitored every 24 h. Results showed that free trypsin maintained only 26% of its initial activity after 5 days, whereas Try@OM-PFC maintained 85% of its initial activity, demonstrating the significantly improved storage stability upon immobilization (Supplementary Fig. 24). Moreover, Try@OM-PFC exhibited remarkable reusability with retaining 80% activity after 4 recycles of reuse (Supplementary Fig. 25), showing its cost-effectiveness and ease of handling advantages for both chemical and proteomic applications. The improved stability and reusability can be ascribed to the existence of hydrogen bonding interaction between trypsin and HOFs, as demonstrated by the shift and new peaks observed in FT-IR spectra. Therefore, OM-PFC exhibit promising potential for loading a variety of biomacromolecules while maintaining active site accessibility and allowing efficient diffusion of substrates and products.

## Try@OM-PFC promoted fibrocyte differentiation

Trypsin has been known to potentiate wound healing for over 50 years, and the 2010 "Federal Guidance for the Use of Medical Preparations" also mentioned its potential use in the wound therapy[39–44]. It had been reported that, when a wound occurs, peripheral blood mononuclear cells (PBMC) leave the blood circulation and enter the wound tissue along with serum. Albumin is the most common protein in serum, accounting for 50% of the total protein in blood. It's found that trypsin can hydrolyze albumin into polypeptide fragments and accelerate cellular nutrient uptake, thus promoting the differentiation of orbicular PBMC into fibrous or fusiform cells called fibrocytes to assist the formation of granulation tissue at the injured site (Fig. 4a)[39]. However, when native trypsin is directly applied to wounds, it is deactivated rapidly due to its autolysis, making this treatment inefficient. Immobilization allows for localizing trypsin in a required region and enhances its stability within the wound environment. From this point of view, immobilized trypsin would be an easy-to-handle, long-acting medication for fibrocyte differentiation.

To assess the effect of Try@OM-PFC on fibrocyte differentiation, PBMC were incubated with the Try@OM-PFC in a medium containing fetal bovine serum (FBS). As shown in Fig. 4 and Supplementary Fig. 26, orbicular cells exhibited enlargement and gradually differentiated into fusiform or fibrous shape in 9 days and accompanied by proliferation. Moreover, Try@OM-PFC21 exhibited a higher quantity of fibrocytes than Try@OM-PFC22 (Fig. 4c, d) and the control group did not show apparent changes. These results suggest that immobilization of trypsin in macroporous HOFs can potentiate fibrocyte differentiation, and the chemical composition of HOFs also influences on the cell proliferation. A plausible reason is that Try@OM-PFC can provide cellular nutrients but avoid the direct contact between trypsin and the wound site (as its high hydrolytic capacity can damage healthy cells and tissues such as skin tissue and blood vessels), therefore reducing potential side effects, burn or local vasodilation[45].

To aid the wound care and the healing process, a suitable wound dressing must be applied at the wound site to protect the injury site from further external mechanical and microbial stress. Therefore, we further prepared a cellular scaffold (Try@OM-PFC/GEL) by mixing Try@OM-PFC with a composite gel scaffold of carboxymethyl cellulose (CMC) and polyethylene glycol (PEG). It was found that the structural integrity of Try@OM-PFC was preserved during the gel-incorporation process (Supplementary Figs. 27 and 28). As shown in Fig. 4e, the resulting cellular scaffold is soft, viscoelastic, enough mechanical strength with the merits of biodegradability and comfort. Its transparent appearance would facilitate the observation of wound conditions, and its viscoelastic nature would allow it to fit snugly on the wound, forming a physical barrier. Moreover, the water-based hydrogel scaffold maintains an optimal hydrated environment. Simultaneously, its porous structure allows the free exchange of nutrients, oxygen and metabolic waste between cells, thus maintaining cell vitality and function. As shown in Supplementary Figs. 29 and 30, Try@OM-PFC/GEL dressing can still hydrolyze BAPNA into yellow PNA, indicating the retained enzymatic activity of Try@OM-PFC within the porous gel matrix and the promising potential for prehospital care.

Herein, we present the synthesis of a series of highly ordered macroporous HOFs in single crystal form based on small biocompatible and clinically applicable molecules through template method. This strategy eliminates the limitation on building blocks for conventional crystalline molecular-based porous material and enable the ease of fabricating large pores in meso- or macro-size, therefore significantly broadening the HOF library for broad applications. Featured with macroporosity, enough stability in a biophysical environment, and excellent biocompatibility, this type of materials serves as promising host matrices for enzyme immobilization and brings improved activity, stability, and recyclability. The obtained Try@OM-PFC potentiates fibrocyte differentiation and can further be fabricated into a cellular scaffold with the merits of biodegradability and comfort for potential applications.

# Methods
## Materials

Styrene (99%), trimesic acid (TMA, 99%), 1, 3, 5-trihydroxybenzene (THB, 99%), uric acid (UA, 98%), methylene blue (MB, 96%), 9,10-diphenylanthracene (DPA, 98%), trypsin (≥4000 U/mg), N-Benzoyl-Arginine-4-nitroanilide (BAPNA, 99%), phosphate buffer saline (PBS, 1×) and carboxymethyl cellulose (CMC, CM-52) were purchased from Adamas. $K_2S_2O_8$ (99%) was purchased from Aldrich. Polyvinyl pyrrolidone (PVP, K30), melamine (MA, 99%), methanol (CH₃OH, 99.5%), polyethylene glycol (PEG, average Mv ~900000,), glycerin (99%) and dichloromethane (CHCl₂, 99.5%) were obtained from Sinopharm. 5(6)-Carboxyfluorescein N-Hydroxysuccinimide Ester (95%) was purchased from MedChemExpress (MCE). Peripheral blood mononuclear cells (PBMCs, isolated from male rats by the supplier) and culture medium were obtained from Pricella Biotechnology Co. Ltd (Wuhan, China; CP-R183). All chemicals were purchased from commercial sources and used without further treatment.

## Synthesis of OM-PFC21

Trimesic acid (TMA, 105 mg, 0.5 mmol) was dissolved in methanol solution (4 mL), and this solution was instilled into the ordered PS template with the diameter of 200 nm and completely adsorbed. The impregnated PS template was dried at 60 °C to obtain TMA@PS. Then TMA@PS was transferred to deionized water (15 mL) with melamine (MA, 63 mg, 0.5 mmol) to trigger the self-assembly of TMA and MA between the interstices of PS spheres. After self-assembly for 24 h, PFC21 was embedded in an ordered PSs template to form PFC21@PS. The obtained PFC21@PS was dispersed in CH₂Cl₂ to remove the PSs, and the product (OM-PFC21) was collected by centrifugation. OM-PFC21 was washed with CH₂Cl₂ several times to ensure the complete removal of PS. Finally, it was washed several times with deionized water and dried (140 mg, yield:83.3%). Other materials were synthesized using similar methods. For details, see the Supplementary Information.

### Synthesis of ordered polystyrene spheres (PSs) template

Monodispersed colloidal PSs with diameters of ~200 nm, 150 nm and 100 nm were synthesized according to a previously reported method[13].

1. 200 nm: the styrene monomer (30 mL) was added to a round-bottomed flask (500 mL) and mixed with water (250 mL) and polyvinyl pyrrolidone (PVP, 0.3 g). After bubbling with nitrogen for 15 min, the mixture was fluxed at 90 °C for 15 min, followed by the addition of aqueous solution (25 mL) containing $K_2S_2O_8$ (0.1 g) as initiator. The polymerization reaction was stirred at 450 rpm for 24 h under $N_2$ atmosphere. After the reaction was cooled, a monodispersed PS emulsion was obtained. Finally, an ordered PS template was obtained by vacuum filtration of the monodispersed PS spheres.

2. 150 nm: the styrene monomer (30 mL) was added to a round-bottomed flask (500 mL) and mixed with water (250 mL) and polyvinyl pyrrolidone (PVP, 0.3 g). After bubbling with nitrogen for 15 min, the mixture was fluxed at 95 °C for 15 min, followed by the addition of aqueous solution (25 mL) containing $K_2S_2O_8$ (0.1 g) as initiator. The polymerization reaction was stirred at 450 rpm for 12 h under $N_2$ atmosphere. After the reaction was cooled, a monodispersed PS emulsion was obtained. Finally, an ordered PS template was obtained by vacuum filtration of the monodispersed PS spheres.

3. 100 nm: the styrene monomer (30 mL) was added to a round-bottomed flask (500 mL) and mixed with water (250 mL) and polyvinyl pyrrolidone (PVP, 0.3 g). After bubbling with nitrogen for 15 min, the mixture was fluxed at 100 °C for 15 min, followed by the addition of aqueous solution (25 mL) containing $K_2S_2O_8$ (0.1 g) as initiator. The polymerization reaction was stirred at 450 rpm for 4 h under $N_2$ atmosphere. After the reaction was cooled, a monodispersed PS emulsion was obtained. Finally, an ordered PS template was obtained by vacuum filtration of the monodispersed PS spheres.

### Synthesis of OM-PFC22

1, 3, 5-trihydroxybenzene (THB, 100 mg, 0.8 mmol) and TMA (168.1 mg, 0.8 mmol) were dissolved in methanol solution (4 mL), and this solution was instilled into the ordered PS template with the diameter of 200 nm and completely adsorbed. The impregnated PS template was dried at 60 °C to form THB + TMA@PS. Subsequently, the THB + TMA@PS was transferred to a mixed solution of methanol and deionized water (v/v = 1:1), triggering the self-assembly of TMA and THB. After self-assembly for 24 h, PFC22 was embedded into the PS monolith to form PFC22@PS. The other procedures were similar with that of OM-PFC21 (183 mg, yield: 83.9%).

### Synthesis of OM-PFC23

Methylene blue (MB, 190 mg, 0.6 mmol) was dissolved in 10 mL methanol solution, and this solution was instilled into the ordered PS template with the diameter of 200 nm and completely adsorbed. The impregnated PS template was dried at 60 °C to obtain MB@PS. Then MB@PS was transferred to 50 mM PBS (50 mL) with uric acid (UA, 100 mg, 0.6 mmol) to trigger the self-assembly of MB and UA (denoted as PFC23). After self-assembly for 24 h, MBU was embedded in an ordered PSs template to form PFC23@PS. The other procedures were similar with that of next step used the same procedure as that for OM-PFC21. OM-PFC23 with macropore diameters of 150 nm and 100 nm was also synthesized by the same process.

### Synthesis of Try@OM-PFC21

Trypsin was dissolved in 50 mM phosphate buffer solution (PBS, pH = 7) to prepare different concentrations ranging from 10 to 80 mg mL$^{-1}$. OM-PFC21 (10 mg) was dispersed in trypsin solution (1 mL), shaken for 24 h to ensure complete adsorption, and centrifuged to collect the Try@OM-

PFC21 product. Then Try@OM-PFC21 was washed with deionized water, and the wash solution contained a negligible amount of free trypsin. The centrifuged solution was collected and the trypsin content in the solution was calculated by the absorbance at 280 nm according to the standard curve of absorbance-concentration.

### Synthesis of Try@OM-PFC22

Try@OM-PFC22 was synthesized using the same procedure as Try@OM-PFC22.

### Synthesis of MA-TMA co-crystal

MA (0.063 g, 0.50 mmol) and TMA (0.105 g, 0.50 mmol) were separately dissolved in hot deionized water (80 mL) to form homogeneous solutions. Subsequently, both solutions were mixed under stirring to initiate self-assembly. The resulting white product (MA-TMA co-crystal) was collected by centrifugation (142 mg, yield: 84.5%).

### Synthesis of THB-TMA co-crystal

TMA (168.1 mg, 0.8 mmol) and THB (100 mg, 0.8 mmol) were dissolved in methanol (8 mL) and mixed to produce THB-TMA co-crystal. Colorless crystals appeared after 2 days. (198 mg, yield: 80.8%).

### Synthesis of fluorescein-labeled trypsin

Trypsin (10 mg) was added to phosphate buffer (10 mL) containing 5(6)-arboxyfluorescein N-Hydroxysuccinimide Ester (0.4 mg) and shocked at 4 °C for 12 h. The resulting fluorescein-labeled trypsin was immobilized in OM-PFC (10 mg) materials.

### Synthesis of Try@OM-PFC/GEL

The matrix of PVP-CMC hydrogel was prepared following the reported literature[46]. A polymer solution consisting of PVP (0.2 g), CMC (0.8 g), PEG (1.0 g), Agar (2.0 g), Glycerin (1 g), and water (95 mL) was used. The polymer solution was processed by using an autoclave to construct three-dimensional porous structures within the hydrogel. Then, Try@OM-PFC was added into the above solution and stirred for 2 min. Finally, the mixture was poured into a round sterile PS petri plate and cooled down to form Try@OM-PFC/GEL.

### Statistics and reproducibility

Data analysis was conducted using the software Microsoft Office PowerPoint 2021 and Origin 2020. Each experiment was independently repeated three times with similar results.

### Reporting summary

Further information on research design is available in the Nature Portfolio Reporting Summary linked to this article.

## Data availability

The experimental data generated in this study are provided within the Article, Supplementary Information and Source Data file. All data are available from the corresponding authors upon request. Source data is available for Fig. 2 and Supplementary Figs. 2–9, 13, 15–27 in the associated source data file. Source data are provided with this paper.

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

## Acknowledgements

The authors gratefully acknowledge the help from Mao-Chun Hong, Yong-Sheng Liu, financial support from the CAS Youth Interdisciplinary Team (grant no. JCTD-2022-12 (T.F.L)), Joint Funds for the Innovation of Science and Technology, Fujian Province (grant no. 2024Y9623 (Z.S.Y)), Science and Technology Program of Fujian Province (grant no. 2025J011174 (Z.S.Y), grant no. 2025J011175 (Z.S.Y)), Fujian Provincial Young and Middle-aged Health Leading Talent Training Program (grant no. 2023-2839 (Z.S.Y)).

## Author contributions

Q.X.L. performed on the design and results analysis of all experiments and wrote the manuscript. W.Z.C. assisted in the cell experiment. X.L.Y. performed TEM characterization. Z.S.Y. and Y.Z. advised on the design and interpretation of cell experiments. A.R.M.S. edited the manuscript. T.F.L. advised on the design and interpretation of all experiments and directed the overall project.

## Competing interests

The authors declare no competing interests.
