## [Transparent Peer Review file · Nature Communications]

Highly Ordered Macroporous Hydrogen-Bonded Organic Frameworks Based on Small Biocompatible Molecules

Corresponding Author: Professor Tian-Fu Liu

Version 0:

Reviewer comments:

Reviewer #1

(Remarks to the Author)

there are several comments that I would like to be addressed first:

-the authors are introducing macroporosity in HOFs by the template method. Although the use of the template method is novel, the introduction of macroporosity is not, as there are several examples in the literature i.e. Angew. Chem. Int. Ed. 2025, e202421523; Angew. Chem. Int. Ed. 2022, 61, e202208677; ACS Appl. Mater. Interfaces 2020, 12, 15588–15594) where some of these papers are using similar building blocks. So they authors should clearly state the advantage of using templates over other methods such as controlling the crystallization via self-assembly beyond the claim of lack of biocompatibility.

The N₂ adsorptions are marginal for some of these compounds.

is THB-TMA crystal novel? How the simulated PXRD was obtained?

the PXRD figures comparing spectra for OM-PFC21, OM-PFC22, OM-PFC23 are too small, and not possible to corroborate the matching. For example, there is a missing peak for OM-PFC23 below 2 θ 10 compared to the simulated spectra, which needs further explanation.

Reviewer #2

(Remarks to the Author)

This manuscript presents a novel and compelling approach by designing highly ordered macroporous hydrogen-bonded organic frameworks (HOFs) based on small biocompatible molecules, which represents a new idea for biomaterial design. The innovative use of HOFs for immobilizing trypsin not only enhances enzymatic stability but also demonstrates remarkable potential in promoting wound healing through fibrocyte differentiation, offering a unique combination of structural precision and biological functionality. The integration of these frameworks into a biocompatible gel scaffold further highlights the practical application potential of this research, making it a highly attractive contribution to the development of new wound care materials. I am glad to recommend its publication in Nature Comm after several questions being addressed:

Q1: In the discussion of "Try@OM-PFC can provide cellular nutrients but avoid the direct contact between trypsin and the wound site", it is recommended to provide a more detailed explanation of why avoiding direct contact reduces potential side effects.

Q2: The manuscript predominantly cites older literature to support "the role of trypsin in promoting wound healing", but fails to mention whether there have been any updated studies in recent years. It is suggested that the authors discuss the latest relevant research or policy to enhance the timeliness of their argument.

Q3: The manuscript raises important questions regarding the specific cellular activities within the Try@OM-PFC/GEL scaffold and the functional role of the scaffold itself. It is crucial to clarify what biological processes the cells undergo within this cell scaffold.

Q4: Could the authors provide a more detailed explanation of the Try@OM-PFC/GEL scaffold's functions, including its mechanical support and facilitation of nutrient and waste exchange, would provide a clearer understanding of its therapeutic potential.

Q5. How does the Try@OM-PFC/GEL scaffold maintain an "optimal hydrated environment"?

Reviewer #3

(Remarks to the Author)

The manuscript by T.-F. Liu et al. presents the synthesis of highly ordered macroporous structures composed of biocompatible hydrogen-bonded organic frameworks (HOFs). These materials were investigated as substrates for trypsin immobilization and as active components in wound-healing patches.

The findings reported by Liu et al. are of significant interest to researchers working with HOFs and will also be highly relevant to the readership of Nature Communications. The synthesis of such highly ordered macroporous HOF-based structures is unprecedented in the literature. From a technological perspective, controlling the 3D structure of a new material is a crucial step toward its practical application. The structural engineering of HOFs remains an open challenge, and this manuscript has the potential to inspire researchers in the field to develop innovative HOF-based materials and devices, ultimately broadening the scope of HOF applications.

The data are clearly presented and strongly support the authors' conclusions. The methodology is also well described. Overall, the manuscript is of high quality and is well suited for publication in Nature Communications.

I do have just a few minor questions regarding the stability of OM-HOFs:

Did the authors quantify the potential leaching of HOF tectons into solution after the soaking tests (Fig. S9)? Did they collect SEM images of the soaked samples?

Similarly, regarding the recyclability tests in Fig. S20: Did the authors attempt to quantify potential HOF degradation or the leaching of trypsin between cycles?

Did the authors directly investigate the structure of the HOF after immobilization in the gel and the experiments shown in Fig. S22? (e.g. XRD or SEM images to demonstrate that the HOF is still crystalline, and the macroporous structure still ordered) In Fig. S22, the time difference between the two images should be explicitly stated in the caption. Additionally, a couple of control experiments could be included here: (i) the same test using an OM-PFC/GEL dressing and (ii) free trypsin immobilized in the GEL.

Reviewer #4

(Remarks to the Author)

In the submitted manuscript, Qiu-Xia Li et al. report the development of biocompatible, molecule-based macroporous materials using a template-assisted approach. While the study presents the generation of macropores within hydrogen-bonded organic frameworks (HOFs), the characterization and analytical details provided are insufficient. The authors are advised to include a comprehensive structural analysis focusing on the template-based synthesis of macroporous HOFs and examine properties such as surface area and thermal stability more extensively. Additionally, the discussion on template-based synthesis is somewhat limited in the 'Results and Discussion' section. An in-depth analysis of the effects of template-based synthesis on the properties of HOFs is necessary. For instance, the manuscript lacks detailed porosity characteristics, and a comparison of how surface area varies with the size of the PS templates could be beneficial.

At this stage, I cannot recommend the manuscript for publication in Nature Communications. I suggest that the authors consider the following points when revising the manuscript:

1. The statement regarding the challenges of introducing macropores, particularly the risks of high toxicity to living organisms due to organic synthesis, is ambiguous (Introduction, Lines 47-49, Page 2): . It is unclear whether the synthetic method, the reagents used, or the final products contribute to these risks. Please clarify this point with appropriate examples.
2. How does the template-based method adopted in this study address the aforementioned toxicity challenges?
3. Given that macropores have been introduced in COFs, MOFs, and organic polymers for enzyme immobilization, what distinct advantages do macroporous HOFs offer over these materials?
4. Please include the size of the PS monoliths and provide the full name of PS upon its first mention.
5. A brief overview of template-based macroporous introduction in HOFs is needed at the end of the introduction section.
6. There are notable differences in the PXRD patterns of OM-PFC23 and PFC23@PS, such as the intense peak at approximately two theta ~19 (Figure S3c). Please discuss the possible reasons for these variations.
7. The term 'nitrogen absorption' should be corrected to 'nitrogen adsorption' (Line 101-102, Page 4).
8. The manuscript describes Type IV isotherms with significant hysteresis loops, typical of mesoporous materials. Please verify the type of adsorption isotherm and provide appropriate references. It is also noted that OM-PFC22 and THB-TMA co-crystal exhibited similar adsorption characteristics. (Line 102-104, Page 4).
9. What are the BET surface areas of these materials?
10. Please use distinct markers or colors for adsorption and desorption curves to enhance clarity and understanding of these patterns.
11. Please specify the non-covalent interactions involved and explain how these contribute to the structural integrity during the template removal process.
12. The explanation of structural details through SAED is too brief. Please expand on this point.
13. What is the stability of the macroporous HOFs in dichloromethane, and how is the weight percentage retained after treatment?
14. Please provide the overall yield comparisons between template-assisted and non-template-based synthesis.
15. Detailed analyses, such as FT-IR, NMR, and TGA, are necessary to ensure the complete removal of the PS template

from the HOFs.

Version 1:

Reviewer comments:

Reviewer #1

(Remarks to the Author)

the authors have addressed my comments. I recommend publication as it is

Reviewer #2

(Remarks to the Author)

The authors have answered the questions raised previously and there are no new ones.

Reviewer #3

(Remarks to the Author)

The authors addressed all the reviewer comments and clarified all the unclear aspects. I think that the manuscript can be published without further revision.

Reviewer #4

(Remarks to the Author)

The authors' response have addressed the referee's comments. I recommend the manuscript for publication after considering the following suggestion: Considering the relatively low BET surface area values of the samples and the inherent limitations in measurement precision, the reported values in the Supporting Information (e.g., 13.22 m²/g) appear overly precise. It is advisable to round such values to reflect the appropriate level of accuracy (e.g., ~13 m²/g), unless the measurement uncertainty is explicitly stated. This will help avoid the impression of unrealistic precision.

Response to reviewers

Reviewer #1 (Remarks to the Author):

Comment 1

1. There are several comments that I would like to be addressed first: the authors are introducing macroporosity in HOFs by the template method. Although the use of the template method is novel, the introduction of macroporosity is not, as there are several examples in the literature i.e. *Angew. Chem. Int. Ed.* 2025, e202421523; *Angew. Chem. Int. Ed.* 2022, 61, e202208677; *ACS Appl. Mater. Interfaces* 2020, 12, 15588-15594) where some of these papers are using similar building blocks. So they authors should clearly state the advantage of using templates over other methods such as controlling the crystallization via self-assembly beyond the claim of lack of biocompatibility.

Response

We thank the reviewer for bringing up these points. We would like to emphasize the advantages of template method from the following aspects:

(1). Regarding to the macroporosity: template method can generate ordered and uniform spherical pores with precisely controllable sizes. In contrast, the methods mentioned in the literatures rely on crystal morphology control to create void space (for example, tuning solvent volatilization rates between crystal faces induces rapid anisotropic self-assembly), which is hard to achieve high-density and uniformly distributed macropores with precise regulation.

(2). Regarding to the building blocks: it is important to highlight that structure in the mentioned literatures are constructed by single monomers, while our hard template strategy allows the construction of macroporous materials using dual or more functional components, and most strikingly, without the restriction on molecular configuration. These features would expand the potential applications scope of these materials.

(3). Regarding to biocompatibility: the template method also affords the flexibility to select building blocks fulfilling the specific requirements for biomedical applications with low toxicity and better biocompatibility.

We deeply appreciate your comments, which has helped us to improve the clarity

and presentation of our findings. We have added a more detailed explanation in the paragraph 2 of page 2 of the manuscript and highlighted in yellow.

Comment 2

2. The N₂ adsorptions are marginal for some of these compounds.

Response

Nitrogen adsorption experiment at 77 K is usually used to evaluate the porosity of materials with micropores (<2 nm) and mesopores (2-50 nm). For the pores falling in this range, nitrogen molecules can interact with the pore wall with strong van der Waals forces, therefore exhibiting high adsorption capacity. However, our OM-HOF and the co-crystalline materials do not contain micropores. The macropores in OM-HOF are significantly larger than nitrogen molecules. As a result, the interaction between nitrogen molecules and the pore surface is very weak, resulting in low nitrogen adsorption capacities. This behavior is consistent with the inherent characteristics of macroporous materials.

Ref: Fernandez A, Hierarchical Assembly of a Micro-and Macroporous Hydrogen-Bonded Organic Framework with Tailored Single-Crystal Size. *Angew. Chem. Int. Ed.* **61**, e202208677 (2022); Alcalde-Santiago V, Three-dimensionally ordered macroporous PrOx: an improved alternative to soot combustion ceria catalysts, *Appl. Catal. B Environ*, (2018).

Comment 3

3. Is THB-TMA crystal novel? How the simulated PXRD was obtained?

the PXRD figures comparing spectra for OM-PFC21, OM-PFC22, OM-PFC23 are too small, and not possible to corroborate the matching. For example, there is a missing peak for OM-PFC23 below 2theta 10 compared to the simulated spectra, which needs further explanation.

Response

We sincerely thank the reviewer for raising this important point.

(1). The THB-TMA crystal has previously been reported (New J. Chem., 2001, 25, 890-

892) and characterized by single-crystal X-ray diffraction. The simulated PXRD patterns were generated using the CIF file provided in this work.

(2). To further address the concern of this reviewer, we replotted the PXRD pattern by reordering and enlarging the curves:

Supplementary Figure 3. The PXRD patterns of (a) PS, PFC21@PS, OM-PFC21 and simulated PFC21, (b) PS, PFC22@PS, OM-PFC22 and simulated PFC22 and (c) PS, PFC23@PS, OM-PFC23 and simulated PFC23.

(3). The missing of some peaks in experimental PXRD patterns can be attributed to the following factors: The use of the PS template affects the crystallite orientation of OM-PFC, which in turn affect the intensity of peaks in the PXRD pattern. Meanwhile, defects and cavities in structure can lead to reduced diffraction intensity and broadened peaks. Due the highly porous structure and low atom density of material, the peak is indeed very weak, but we can still find the characteristic peak attributing to THB-TMA, MA-TMA, and MBU. Moreover, the high-resolution TEM confirmed the consistent crystal structures between OM-PFC23 and MBU.

Ref: Sandesh Jadkar, Stabilizing TiO₂/CsPbI₂Br Perovskite Buried Interface for All-Inorganic Perovskite toward Highly Efficient Photodetectors. *Adv. Mater. Interfaces*, 2500052 (2025); Roger Smith, Influence of temperature and point defects on the X-ray diffraction pattern of graphite. *Carbon Trends*. **5**, 100124 (2021).

Reviewer #2 (Remarks to the Author):

This manuscript presents a novel and compelling approach by designing highly ordered macroporous hydrogen-bonded organic frameworks (HOFs) based on small biocompatible molecules, which represents a new idea for biomaterial design. The innovative use of HOFs for immobilizing trypsin not only enhances enzymatic stability but also demonstrates remarkable potential in promoting wound healing through fibrocyte differentiation, offering a unique combination of structural precision and biological functionality. The integration of these frameworks into a biocompatible gel scaffold further highlights the practical application potential of this research, making it a highly attractive contribution to the development of new wound care materials. I am glad to recommend its publication in Nature Comm after several questions being addressed:

Response

We thank the reviewer for the positive comments. Please find below our point-by-point responses to your concerns.

Comment 1

1. In the discussion of "Try@OM-PFC can provide cellular nutrients but avoid the direct contact between trypsin and the wound site", it is recommended to provide a more detailed explanation of why avoiding direct contact reduces potential side effects.

Response

We appreciate the reviewer's insightful comments. We have added a more detailed explanation on the mechanism: The Try@OM-PFC minimizes direct trypsin-wound contact, as its high hydrolytic capacity can damage healthy cells and tissues (e.g., skin tissue and blood vessels) and avoiding side effects such as pain, hyperemia, or edema. This design not only enhances the safety of the treatment but also strengthens its potential for clinical application. We have added more explanations in main text and highlighted by yellow.

Ref: Laura KS Parnell, Wound dressing components degrade proteins detrimental

to wound healing, *Int Wound J.* **5**, 543-51 (2008); Proteases and Delayed Wound Healing, *Adv Wound Care* (New Rochelle). **2**, 438-447 (2013); International consensus. The role of proteases in wound diagnostics. An expert working group review. London: Wounds International, (2011).

Comment 2

2. The manuscript predominantly cites older literature to support “the role of trypsin in promoting wound healing”, but fails to mention whether there have been any updated studies in recent years. It is suggested that the authors discuss the latest relevant research or policy to enhance the timeliness of their argument.

Response

We sincerely thank the reviewer for this valuable suggestion. We have revised our manuscript to include the latest research. As follows:

44 Hu, D. *et al.* Accelerated healing of intractable biofilm-infected diabetic wounds by trypsin-loaded quaternized chitosan hydrogels that disrupt extracellular polymeric substances and eradicate bacteria. *Int. J. Biol. Macromol.* **278** (2024).

45 Xiang, Y., Jiang, Y. & Lu, L. Low-Dose Trypsin Accelerates Wound Healing via Protease-Activated Receptor 2. *ACS Pharmacol Transl Sci.* **7**, 274-284 (2024).

Comment 3

3. The manuscript raises important questions regarding the specific cellular activities within the Try@OM-PFC/GEL scaffold and the functional role of the scaffold itself. It is crucial to clarify what biological processes the cells undergo within this cell scaffold.

Response

We appreciate the reviewer's insightful comment. We provide a more detailed explanation:

Scaffold can provide support for cell adhesion, proliferation, migration, differentiation and metabolism. For example, the scaffold can provide a microenvironment conducive to promote PBMC-to-fibrocyte differentiation by hydrolyzing albumin into polypeptide fragments, thereby accelerating cellular nutrient

uptake and the formation of granulation tissue at the injured site. Simultaneously, its porous structure allows the free exchange of nutrients, oxygen and metabolic waste between cells, thus maintaining cell vitality and function.

Comment 4

4. Could the authors provide a more detailed explanation of the Try@OM-PFC/GEL scaffold's functions, including its mechanical support and facilitation of nutrient and waste exchange, would provide a clearer understanding of its therapeutic potential.

Response

We thank the reviewer for this valuable suggestion. We provide a more detailed explanation:

(1). Mechanical Support: The scaffold features a biomimetic 3D porous architecture that mimics the natural extracellular matrix, providing optimal structural support for cell attachment and tissue regeneration. The combination of carboxymethyl cellulose and polyethylene glycol in the scaffold ensures sufficient mechanical strength to endure physiological stresses (e.g., stretching or compression) while maintaining flexibility to adapt to wound.

(2). Facilitation of Nutrient and Waste Exchange: The scaffold's interconnected porous structure enables efficient exchange of oxygen, nutrients, and metabolic waste, maintaining cellular viability. Meanwhile, the trypsin within scaffold actively degrades wound debris (e.g., clot proteins) to cleanse the wound bed while releasing bioactive peptides that support healing. The above promotes tissue repair by sustaining cellular metabolism and creating a favorable wound microenvironment.

Comment 5

5. How does the Try@OM-PFC/GEL scaffold maintain an "optimal hydrated environment"?

Response

We thank the reviewer for bringing up this important point. We can provide the following response:

The scaffold with carboxymethyl cellulose and polyethylene glycol is typically

composed of hydrophilic polymers, which exhibit excellent water retention capabilities. Its porous structure dynamically regulates moisture balance by absorbing excess exudate or releasing moisture as needed, while simultaneously serving as a vapor barrier to prevent desiccation.

Ref: Feng S.M., Research progress of skin tissue engineering scaffolds and their materials in wound repair, *Med J PUMCH*, **14**, 603-610 (2023).

Reviewer #3 (Remarks to the Author):

The manuscript by T.-F. Liu et al. presents the synthesis of highly ordered macroporous structures composed of biocompatible hydrogen-bonded organic frameworks (HOFs). These materials were investigated as substrates for trypsin immobilization and as active components in wound-healing patches.

The findings reported by Liu et al. are of significant interest to researchers working with HOFs and will also be highly relevant to the readership of Nature Communications. The synthesis of such highly ordered macroporous HOF-based structures is unprecedented in the literature. From a technological perspective, controlling the 3D structure of a new material is a crucial step toward its practical application. The structural engineering of HOFs remains an open challenge, and this manuscript has the potential to inspire researchers in the field to develop innovative HOF-based materials and devices, ultimately broadening the scope of HOF applications.

The data are clearly presented and strongly support the authors' conclusions. The methodology is also well described. Overall, the manuscript is of high quality and is well suited for publication in Nature Communications.

I do have just a few minor questions regarding the stability of OM-HOFs:

Response

We thank the reviewer for the positive feedbacks and the careful review of our manuscript. Please find our point-by-point response to your comments below.

Comment 1

1. Did the authors quantify the potential leaching of HOF tectons into solution after the soaking tests (Fig. S9)?

Response

We appreciate the reviewer's insightful question. To address this concern, we monitored the mass change before and after the soaking process. The results show that only a small amount of mass change was observed, which can be attributed to the losses brought by centrifugation. These data have been added to **Supplementary Figure 13**

for clarity.

Supplementary Figure 13. The PXR D patterns of (a) OM-PFC21 and (b) OM-PFC22 soaked in 50 mM PBS and artificial sweat for a week. (c) The mass retention ratio of the OM-PFC21 and OM-PFC22 before and after soaking in 50 mM PBS, artificial sweat and CH₂Cl₂ for a week.

Comment 2

2. Did they collect SEM images of the soaked samples?

Response

We thank the reviewer for this suggestion. SEM images of the soaked samples (in PBS and artificial sweat) have now been added as **Supplementary Figure 14** in the Supporting Information. The images confirm that the morphologies of OM-PFC21 and 22 remain intact after soaking, demonstrating their structural stability under these conditions.

Supplementary Figure 14. SEM images of (a-b) OM-PFC21 and (c-d) OM-PFC22 soaked in 50 mM PBS and artificial sweat for a week.

Comment 3

3. Similarly, regarding the recyclability tests in Fig. S20: Did the authors attempt to quantify potential HOF degradation or the leaching of trypsin between cycles?

Response

We sincerely appreciate the reviewer's insightful question. Additional characterization was performed to quantify both Try@OM-PFC integrity and trypsin leaching during recycling. As shown in **Supplementary Figure 25**, we monitored the mass change of Try@OM-PFC for each cycle and the UV-Vis spectra of its supernatant after BAPNA hydrolysis. The resulting analysis was added in the caption of **Supplementary Figure 25**.

Supplementary Figure 25. The catalytic activity of recycled (a) Try@OM-PFC21 and (b) Try@OM-PFC22. (c) The mass retention of recycled Try@OM-PFC21 and 22, and the UV-vis of (d) Try@OM-PFC21 and (e) Try@OM-PFC22 supernatant. The results indicated a 10~20% weight loss after four recycles caused by the centrifugation operation and resulted in a slight decrease in catalytic activity. UV-Vis spectra of the supernatant showed no significant absorption peak at 280 nm, the characteristic absorption of trypsin, confirming no enzyme leaching. The above results demonstrate the recyclability of Try@OM-PFC.

Comment 4

4. Did the authors directly investigate the structure of the HOF after immobilization in the gel and the experiments shown in Fig. S22? (e.g. XRD or SEM images to demonstrate that the HOF is still crystalline, and the macroporous structure still ordered).

Response

As suggested, we freeze-dried the Try@OM-PFC/GEL composite and then conducted XRD analysis and SEM characterization (now shown in the **Supplementary Figure 27** and **28**). The XRD patterns confirm that the Try@OM-PFC maintains its crystallinity within the gel matrix, as evidenced by the consistent characteristic peaks between Try@OM-PFC/GEL and the co-crystal structure (**Supplementary Figure 27**).

SEM images (**Supplementary Figure 28**) further demonstrate that Try@OM-PFC remains ordered macroporous structure after immobilization. These results strongly support that the structural integrity is preserved during the gel-incorporation process. We have also added related discussion in the main text and highlighted in yellow.

Supplementary Figure 27. The PXR D patterns of (a) Try@OM-PFC21/GEL and (b) Try@OM-PFC22/GEL after freeze-drying.

Supplementary Figure 28. SEM images of Try@OM-PFC/GEL after freeze-drying.

Comment 5

5. In Fig. S22, the time difference between the two images should be explicitly stated in the caption. Additionally, a couple of control experiments could be included here: (i)

the same test using an OM-PFC/GEL dressing and (ii) free trypsin immobilized in the GEL.

Response

We sincerely appreciate the reviewer's insightful suggestions, which are very valuable for improving our manuscript. According to the comments, we have now included the time interval between these images and added control experiments as shown in **Supplementary Figure 30**:

Supplementary Figure 30. Time-dependent hydrolysis of BAPNA by (i) Try/GEL, (ii) Try@OM-PFC /GEL, and (iii) OM-PFC/GEL control dressings at (a) 10, (b) 15, and (c) 30 min incubation intervals. The above results show that the Try/GEL system demonstrates faster BAPNA-to-PNA hydrolysis rate due to the easily access to catalytic active sites of free trypsin. OM-PFC/GEL without trypsin (negative control) shows no hydrolytic activity, confirming that the catalytic activity is derived from the enzyme.

Reviewer #4 (Remarks to the Author):

In the submitted manuscript, Qiu-Xia Li et al. report the development of biocompatible, molecule-based macroporous materials using a template-assisted approach. While the study presents the generation of macropores within hydrogen-bonded organic frameworks (HOFs), the characterization and analytical details provided are insufficient. The authors are advised to include a comprehensive structural analysis focusing on the template-based synthesis of macroporous HOFs and examine properties such as surface area and thermal stability more extensively. Additionally, the discussion on template-based synthesis is somewhat limited in the 'Results and Discussion' section. An in-depth analysis of the effects of template-based synthesis on the properties of HOFs is necessary. For instance, the manuscript lacks detailed porosity characteristics, and a comparison of how surface area varies with the size of the PS templates could be beneficial.

At this stage, I cannot recommend the manuscript for publication in Nature Communications. I suggest that the authors consider the following points when revising the manuscript:

Response

We sincerely appreciate the reviewer's valuable suggestions for improving our manuscript. We have carefully considered all comments and are committed to implementing substantial revisions to address each concern. We provide a detailed, point-by-point response to these questions below:

Comment 1

1. The statement regarding the challenges of introducing macropores, particularly the risks of high toxicity to living organisms due to organic synthesis, is ambiguous (Introduction, Lines 47-49, Page 2): It is unclear whether the synthetic method, the reagents used, or the final products contribute to these risks. Please clarify this point with appropriate examples.

Response

We appreciate the reviewer's comment. The more detailed explanation of the toxicity risks of macroporous structure with large-size organic building blocks is listed below:

First, large-size organic building blocks often possess potential intrinsic toxicity. Their complex molecular structures and the presence of various functional groups can lead to adverse interactions with biological systems, such as binding to critical biomolecules or disrupting cellular processes. Second, the degradation difficulty of these compounds further exacerbates their toxicity. Due to their large and complex structures, they are often resistant to metabolic pathways and degradation processes. As a result, these compounds accumulate in living organisms over time, increasing the likelihood of long-term physiological harm. Therefore, careful building block selection and synthetic strategies are crucial to mitigate these risks in biomedical applications.

Comment 2

2. How does the template-based method adopted in this study address the aforementioned toxicity challenges?

Response

We appreciate the reviewer's question. The detailed explanation is listed below:

In this study, we used low-toxicity organic building blocks, including melamine (LD₅₀: 4550 mg/kg), tricarboxylic acid (LD₅₀: 16000 mg/kg), m-trihydroxybenzene (LD₅₀: 5200 mg/kg), methylene blue (LD₅₀: 3500 mg/kg), and uric acid (LD₅₀: 5040 mg/kg). These compounds are either clinically approved or have very low physiological toxicity. Moreover, the synthesis is conducted with water and ethanol as solvents, which are environmentally friendly and non-toxic compared to the solvents usually used for organic linker synthesis. The above features minimize the risk of toxic residues in the final product.

These characteristics ensure that the resulting macroporous materials are low-toxicity and suitable for biomedical applications.

Comment 3

3. Given that macropores have been introduced in COFs, MOFs, and organic polymers for enzyme immobilization, what distinct advantages do macroporous HOFs offer over these materials?

Response

We thank the reviewer for this insightful question. The benefits of macroporous HOFs are listed below:

OM-HOFs exhibit inherent biocompatibility and low toxicity due to their metal-free components (compared with MOFs) and the assembly based on non-covalent interactions (compared with COFs), as well as the use of clinically approved or low-toxicity building blocks. Moreover, the synthesis of OM-HOFs underwent in mild conditions, avoiding the use of high temperatures, strong acids/bases, or toxic solvents. These advantages make OM-HOFs particularly suitable for enzyme immobilization in biological applications.

Comment 4

4. Please include the size of the PS monoliths and provide the full name of PS upon its first mention.

Response

We thank the reviewer for this valuable suggestion. We have added the size of the polystyrene (PS) monoliths and provided the full name of PS upon its first mention, which is highlighted in yellow in the revised manuscript.

Comment 5

5. A brief overview of template-based macroporous introduction in HOFs is needed at the end of the introduction section.

Response

We appreciate the reviewer's suggestion. We provide a brief overview at the end of the introduction section, highlighted in yellow in the revised manuscript:

For these reasons, current strategies for fabricating macroporous materials based

on NCI assembly, such as hydrogen-bonded organic frameworks (HOFs), have largely relied on crystal morphology control to create void spaces.²³⁻²⁵ For example, tuning solvent volatilization rates induced rapid anisotropic self-assembly to generate macropores. However, a broadly applicable strategy for producing HOFs with high-density and uniformly macropores has yet to be established.

23 Cui, J.-W., Liu, S.-H., Tan, L.-X. & Sun, J.-K. Engineering Hierarchy to Porous Organic Cages for Biomimetic Catalytic Applications. *Acc. Mater. Res.* (2025).

24 Halliwell, C. A. *et al.* Hierarchical Assembly of a Micro- and Macroporous Hydrogen-Bonded Organic Framework with Tailored Single-Crystal Size. *Angew. Chem. Int. Ed.* **61**, e202208677 (2022).

25 Tothadi, S., Koner, K., Dey, K., Addicoat, M. & Banerjee, R. Morphological Evolution of Two-Dimensional Porous Hexagonal Trimesic Acid Framework. *ACS Appl. Mater. Interfaces.* **12**, 15588-15594 (2020).

Comment 6

6. There are notable differences in the PXRD patterns of OM-PFC23 and PFC23@PS, such as the intense peak at approximately two theta ~19 (Figure S3c). Please discuss the possible reasons for these variations.

Response

If we zoom in the region of the 2θ range between 18 to 24, there is a small peak but with very low intensity, which can be attributed to the change of crystallite orientation. The presence of the PS template and macroporous defect affects the crystallite orientation, resulting in the increased or decreased intensity of peaks in the PXRD pattern.

Comment 7

7. The term 'nitrogen absorption' should be corrected to 'nitrogen adsorption' (Line 101-102, Page 4).

Response

We sincerely thank the reviewer. The "nitrogen absorption" has been corrected to "nitrogen adsorption", highlighted in yellow in the revised manuscript.

Comment 8

8. The manuscript describes Type IV isotherms with significant hysteresis loops, typical of mesoporous materials. Please verify the type of adsorption isotherm and provide appropriate references. It is also noted that OM-PFC22 and THB-TMA co-crystal exhibited similar adsorption characteristics. (Line 102-104, Page 4).

Response

We thank the reviewer for the valuable comments.

The above figure shows four classic Type IV isotherm hysteresis curves (*Surf. Interface Anal.* **36**, 1323-1329 (2004)). According to the International Union of Pure and Applied Chemistry (IUPAC) guidelines, Type IV isotherms are defined by a hysteresis loop occurring and a sharp increase at relatively high pressures, indicating capillary condensation within mesopores and macropores. This description is consistent with the adsorption isotherms of the OM-HOF materials. To support this conclusion, we have added references on adsorption isotherms: 1. Sing, K. S. W. et al., Reporting physisorption data for gas/solid systems with special reference to the determination of surface area and porosity. *Pure and Applied Chemistry*, **57**, 603-619 (1985); 2. E. Lester, N. Hilal and J. Henderson, Porosity in ancient glass from Syria (c. 800 AD) using gas adsorption and atomic force microscopy. *Surf. Interface Anal.* **36**, 1323-1329 (2004).

(2). OM-PFC22 and THB-TMA co-crystal exhibited different adsorption isotherms. At high relative pressures ($P/P_0 \approx 1$), the adsorption curve of OM-PFC22 shows a sharp increase, suggesting the presence of large pores. This is because capillary condensation of large pores requires a high relative pressure to occur. When the relative pressure P/P_0 is close to 1, nitrogen in the large pores begins to condense, resulting in a sharp increase in adsorption capacity (ref: Lowell, S., Shields, J. E., Thomas, M. A., & Thommes, M. *Characterization of Porous Solids and Powders: Surface Area, Pore*

Size and Density. *Particle Technology Series* (2004). doi:10.1007/978-1-4020-2303-3.).

In contrast, this sharp increase is absent in the isotherm of THB-TMA co-crystal.

Comment 9

9. What are the BET surface areas of these materials?

Response

The BET surface area of OM-PFC21 is 13.22 m²/g; MA-TMA co-crystal is 6.06 m²/g; OM-PFC22 is 2.39 m²/g; THB-TMA co-crystal is 1.72 m²/g. This information was added in the caption of **Supplementary Figure 4**.

Comment 10

10. Please use distinct markers or colors for adsorption and desorption curves to enhance clarity and understanding of these patterns.

Response

Thank you for the valuable suggestion. We have revised the **Supplementary Figure 4** to distinct the adsorption and desorption curves:

Supplementary Figure 4. The N₂ adsorption-desorption isotherms measured at 77 K (a) OM-PFC21 and MA-TMA co-crystal, (b) OM-PFC22 and THB-TMA co-crystal. The BET surface area is 13.22 m²/g for OM-PFC21, 6.06 m²/g for MA-TMA co-crystal, 2.39 m²/g for OM-PFC22, 1.72 m²/g for THB-TMA co-crystal.

Comment 11

11. Please specify the non-covalent interactions involved and explain how these contribute to the structural integrity during the template removal process.

Response

Thank you for the insightful question. We can imagine that increased molecular packing leads to a greater number of non-covalent interactions between molecules, resulting in a more stable structure. Using small molecules as building blocks means there exists more molecules in a unit volume, therefore leading to high-density and stronger NCIs, thereby significantly enhances structural and solvent stability during template removal. Among them, abundant hydrogen bonding provides directional stability, maintaining the overall architecture. π - π stacking and Van der Waals forces offer additional interactions, ensuring the framework stability withstanding the stresses of template removal.

Comment 12

12. The explanation of structural details through SAED is too brief. Please expand on this point.

Response

We have added the detailed explanation as below:

In this study, the SAED patterns exhibited sharp and well-defined diffraction spots, confirming the high crystallinity of the material. The lattice parameters measured from the ring/spot radius of the SAED patterns were consistent with the parameters derived from single-crystal X-ray diffraction (SCXRD) data, validating the consistency of the material structure with single-crystal structure. Additionally, the SAED patterns revealed a preferred crystal orientation along the [001] and [111] crystallographic planes, which aligns with the observed PXRD growth direction. The absence of additional diffraction rings or spots indicated that the sample was free of another phases or impurities.

Comment 13

13. What is the stability of the macroporous HOFs in dichloromethane, and how is the

weight percentage retained after treatment?

Response

To evaluate the stability, we immersed the macroporous HOFs in dichloromethane for 7 days at room temperature. The samples were centrifuged and vacuum dried at 60°C for 12 h. The recovered material retained about 97% of its original weight, demonstrating its decent stability in dichloromethane. (**Supplementary Figure 13c** of the revised supporting information).

Comment 14

14. Please provide the overall yield comparisons between template-assisted and non-template-based synthesis.

Response

OM-PFC21: 140 mg, yield:83.3%; MA-TMA co-crystal: 142 mg, yield: 84.5%; OM-PFC22: 183 mg, yield: 83.9%; THB-TMA co-crystal: 198 mg, yield: 80.8%. The yield of template-assisted synthesis is close to that of non-template-assisted synthesis. We added this information at the end of the synthesis procedure and highlighted in yellow.

Comment 15

15. Detailed analyses, such as FT-IR, NMR, and TGA, are necessary to ensure the complete removal of the PS template from the HOFs.

Response

Thanks for your comments sincerely. We have performed NMR, FTIR, and TGA analyses. These results confirm the complete removal of the PS template from the HOFs, we added more detailed discussion in **Supplementary Figure 5-8** of the revised Supporting Information.

Supplementary Figure 5. The FT-IR Spectra of (a) OM-PFC21, PS and PFC21@PS, (b) OM-PFC22, PS, and PFC22@PS. Characteristic peak of the PS template was not observed in spectrum of OM-PFC21 and 22, indicating the complete template removal.

Supplementary Figure 6. ^1H NMR spectra of (a) PFC21@PS, (b) PFC22@PS and (c) PS. These composites were soaked in deuterated chloroform for test after the insoluble solids being filtered out.

Supplementary Figure 7. ^1H NMR spectra of (a) OM-PFC21 and (b) OM-PFC22. OM-PFC was immersed in deuterated chloroform solution. No peaks derived from polystyrene were found, indicating the complete template removal.

Supplementary Figure 8. Thermo-gravimetric Analysis (TGA) curves of OM-PFC21, PS and PFC21@PS, (b) OM-PFC22, PS and PFC22@PS.

Response to reviewers

Title: Highly Ordered Macroporous Hydrogen-Bonded Organic Frameworks Based on Small Biocompatible Molecules

Manuscript ID: NCOMMS-25-05409

Reviewer #1:

Comment

The authors have addressed my comments. I recommend publication as it is

Response

We thank this reviewer for the positive feedback.

Reviewer #2:

Comment

The authors have answered the questions raised previously and there are no new ones.

Response

We thank this reviewer for the positive feedback.

Reviewer #3:

Comment

The authors addressed all the reviewer comments and clarified all the unclear aspects.

I think that the manuscript can be published without further revision.

Response

We thank this reviewer for the positive feedback.

Reviewer #4

Comment

The authors' response have addressed the referee's comments. I recommend the manuscript for publication after considering the following suggestion: Considering the relatively low BET surface area values of the samples and the inherent limitations in

measurement precision, the reported values in the Supporting Information (e.g., 13.22 m²/g) appear overly precise. It is advisable to round such values to reflect the appropriate level of accuracy (e.g., ~13 m²/g), unless the measurement uncertainty is explicitly stated. This will help avoid the impression of unrealistic precision.

Response

We thank this reviewer for the valuable suggestion. We have revised it.